

# On-line SPME derivatization for the sensitive determination of multi-oxygenated volatile compounds in air

Esther Borrás[1], Luis A. Tortajada-Genaro[2], Milagros Ródenas[1], Teresa Vera[1], Thomas Speak[3], Paul

Seakins[3], Marvin D. Shaw[4], Alastair C. Lewis[4], Amalia Muñoz[1] [*]

[1] Fundación Centro de Estudios Ambientales del Mediterráneo (CEAM), 46980 Paterna, Valencia, Spain
[2] Departamento de Química-Instituto IDM. Universitat Politècnica de València, 46022, Valencia, Spain.
[3] University of Leeds, LS2 9JT, Leeds, UK.
[4] National Centre for Atmospheric Science, University of York, York YO10 5DD, UK.

*Correspondence to*: Amalia Muñoz (amalia@ceam.es)






**Abstract.** Multi-oxygenated volatile organic compounds are important markers of air pollution and precursors of ozone and secondary aerosols in both polluted and remote environments. Herein, their accurate determination was enhanced. The approach was based on an automated system for active sampling and on-fiber derivatization coupled with GC-MS technique. The method capability was determined for different compound families, such as aldehydes, ketones, α-dicarbonyls, hydroxy-

aldehydes, hydroxy-ketones and, carboxylic acids. A good accuracy (<7%) was demonstrated from the results compared to Fourier-transform infrared spectroscopy (FTIR). Detection limits of 6 - 100 pptV were achieved with a time resolution lower than 20 min. The developed method was successfully applied to the determination of multi-oxygenated compounds in air samples collected during an intercomparison campaign (EUROCHAMP-2020 project). Also, its capability and accuracy for atmospheric monitoring was demonstrated in an isoprene ozonolysis experiment. Both were carried out in the high-volume

outdoor atmospheric simulation chambers (EUPHORE, 200 m$^3$).

In summary, our developed technique offers near-real-time monitoring with direct sampling, which is an advantage in terms of handling and labour time for a proper quantification of trace-levels of atmospheric multi-oxygenated compounds.




# 1 Introduction

The interest in the assessment of multi-oxygenated volatile organic compounds, so called OVOCs, has increased significantly in recent years. These compounds exist naturally in the environment but mainly come from the photo-oxidation of hydrocarbons in the atmosphere (Mellouki et al., 2015). They play an important role in secondary organic aerosol (SOA) and ozone formation, can potentially influence climate change and cause negative health effects on humans (Atkinson and Arey, 2003). Their tropospheric range levels are highly variable, generally at trace levels, thus adding more complexity to their determination (Gomez-Alvarez et al., 2012). Moreover, OVOCs are polar and volatile, a fact that negatively affects their determination. Given their crucial role in tropospheric chemistry, there is an urgent need for sensitive and reliable techniques.

Several technologies have been proposed for the determination of OVOCs, including Fourier transform infrared spectroscopy (FTIR), proton transfer mass spectrometry (PTR-MS), selected-ion flow-tube mass spectrometry (SIFT-MS), broadband cavity enhanced absorption spectroscopy (BBCEAS), cavity-enhanced differential optical absorption spectroscopy (CE-DOAS) (Thalman et al., 2015). Atmospheric pressure interface time-of-flight chemical ionization mass spectrometers (APITOF-CIMS) has been recently proposed for the detection of OVOCs. Nevertheless, CIMS data present overlapping ions and large uncertainties in quantification of the wide variety of species have been detected (Riva et al., 2019). Moreover, some off-line methods, generally based on chromatographic techniques, have been applied (Legreid et al., 2007). They incorporate intensive sample pre-concentration steps by passing through high volumes of air. The sampling systems are denuders, bags, lab-on-chip devices, impingers and different types of solid-phase cartridges (Ras et al., 2009). In the recent years, solid phase microextraction (SPME) has demonstrated an extraordinary potential for the sampling of volatile compounds, supporting a more accurate determination of multi-oxygenated compounds (Zhu et al., 2015). The advantages of SPME are low cost per sample, reusability, high selective sampling for target analytes, high sensitivity, without solvent extraction - meeting the requirement of green chemistry -, high reproducibility, quite fast and it can be automated (Chen and Pawliszyn, 2004; Gómez-Alvarez, 2007; Baimatova et al., 2016).

Because of high volatility and polarity of OVOCs, their latter quantification is quite complex and usually it is necessary to derivatize prior to chromatographic analysis. Derivatization enhances chromatographic behaviour or detectability and it enables a resolved separation of species which are not directly amenable to analysis due to inadequate volatility or stability (Edler et al., 2002). The main reagents for carbonyl determination include dinitrophenyl hydrazine (DNPH) – normally used in conjunction with high-performance liquid chromatography (HPLC) analysis – (Van Leeuwen et al., 2004) and O-(2,3,4,5,6-pentafluorophenyl) methylhydroxylamine (PFBHA) that is usually applied in conjunction with gas chromatography (GC) (Yu et al., 1995). Silylant reagents such as N,O-bis(trimethylsilyl)trifluoroacetamide (BSTFA) or N-methyl-n-(trimethylsilyl) trifluoroacetamide (MSTFA) were employed for hydroxyl or carboxylic functional groups derivatization (Borras and Tortajada-Genaro, 2012; Jaoui et al., 2012). These types of derivatization are performed after sampling and are followed by lengthy and aggressive sample treatment, which could potentially alter the sample



composition. Moreover, some of them were questioned due to interferences with water vapor, ozone or nitrogen dioxide (Mellouki et al., 2015).

The key to making major improvements in derivatization methodologies may lie in performing simple and automated sample preparation prior to analysis (Pang et al., 2013). In this way, PFBHA derivatization combined with SPME provides several

exceptional advantages for carbonyl compounds (Bourdin and Desauziers 2014). After the sampling, derivatization takes place immediately on-fiber and the derivatives are thermally desorbed, avoiding dilution by solvents and sample alteration. However, previous studies were carried out at concentrations of analytes considerably higher (hundreds of ppbV) than those that would be present in atmospheric conditions. In addition, the decrease of fiber efficiency due to deterioration or depletion of the polymeric phase, was also observed. Another problem is the competition among analytes and the consequent reduction

of reaction yields (Larroque et al., 2006).

These facts have led us (1) to explore on-line variations in the SPME derivatization which could lead to improving its performance and (2) to conduct experiments aimed at evaluating the quantification approach for atmospheric studies. Therefore, a method based on active sampling with SPME fibers and quantitative GC-MS determination of OVOCs had been developed. The research aim was to investigate on-line derivatization for decreasing the sampling times, reaching an

effective capture and a complete conversion, and reducing competition between analytes. In order to evaluate the performances, the simultaneous determination of OVOCs in air samples was examined at the EUPHORE chamber facility, a highly instrumented, large-scale outdoor simulation chamber (Borras et al., 2015).

## 2 Experimental

### 2.1 Reagents

The derivatization reagents: O-(2,3,4,5,6- pentafluorobenzyl) hydroxylamine (PFBHA) (99%); N,O-bis(trimethylsilyl)trifluoroacetamide (BSTFA), N-trimethylsilyl-N-methyl trifluoroacetamide (MSTFA) and trimethylchlorosilane (TMCs) as catalyst purchased from Sigma–Aldrich (Madrid, Spain) were used directly in this study without any further purification. Methylglyoxal at 30% in water (MGLY), glyoxal (GLY), glutaraldehyde, methyl vinyl ketone (MVK), methacrolein, glycoladehyde, hydroxyacetone, succinic acid and benzaldehyde were supplied by Fluka

(Madrid, Spain). Other α-dicarbonyl compounds, such as 4-oxo-2-pentenal and E-butendial, were synthetized (provided by Leeds University, Organic Chemistry Department).

### 2.2 Automated active SPME sampling coupled to GC-MS system

This methodology was carried out in four steps. The first one was the chemical modification of the fiber with PFBHA

(carbonyl reagent). The second proposed step was the sampling on a cell of multi-oxygenated compounds for "on-fiber derivatization" of carbonyl compounds. The third step was the chemical modification of the fiber with MSTFA plus TMCs (hydroxyl or carboxylic compounds reagent). Finally, derivatized OVOCs on SPME were injected on GC for thermal



desorption and their analysis by mass spectrometry. These steps were carried out by an automated procedure achieved with a Combi Pal autosampler (CTC Analytics, Zwingen, Switzerland) controlled by Cycle Composer software and equipped with

SPME sampling adaptors, sample trays, and a temperature-controlled agitator tray. Figure 1 shows a scheme of multi-step derivatization protocol.

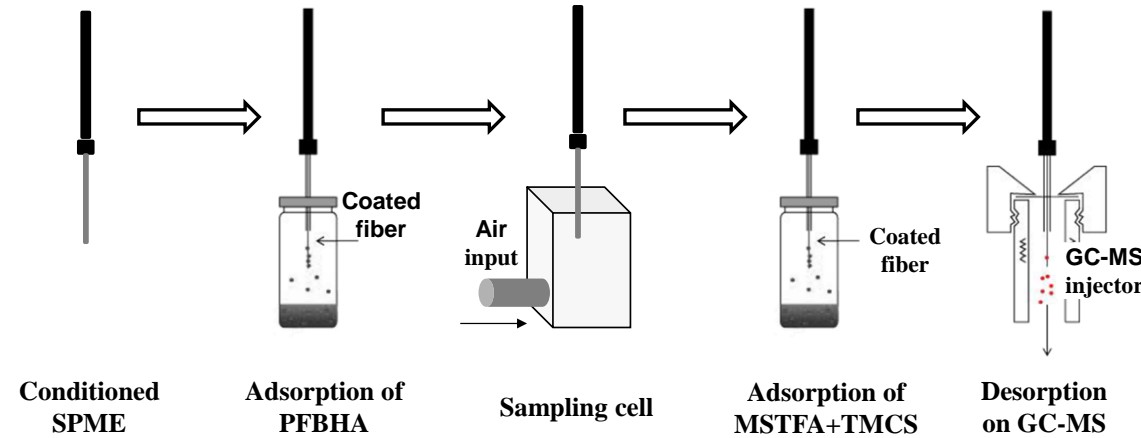

Fig.1 Scheme of sampling and on-fiber derivatization steps.

For the development of our proposed methodology, a sampling system with an inert sampling line and a cell was designed. Sampling line was of sulfinert® material and it was heated at 80ºC to avoid losses of steaky OVOCs compounds. The sampling cell of 5 x 4 x 4 cm was made of stainless steel. The air passed through, in a turbulent regime at 10 L min$^{-1}$, which corresponds to 50 cm s$^{-1}$, to guarantee air velocity higher than 10 cm s$^{-1}$, defined by some authors as a certain critical value (Gómez Alvarez et al., 2012; Augusto et al., 2001). Moreover, the sampling cell was installed just below the simulator

chamber but at laboratory conditions, i.e. without the influence of solar radiation and under controlled temperature, critical parameters for SPME sampling (Gomez-Alvarez, 2007).

The SPME selected fiber was polydimethylsiloxane/divinylbenzene (PDMS/DVB) coating, a stable flex fiber, with a needle size 23 gauge and, coating thickness of 65 µm (Sigma Aldrich, USA). This type of fiber was chosen because of its high affinity to PFBHA (Zhu et al., 2015). A robotic injection system (PAL Auto Sampler, Agilent Technologies, USA) was used.

The protocol started with the conditioning of SPME fiber in the GC injector port (30 min at 200ºC). By doing so, every analysis also constitutes pre-conditioning for the following sampling step avoiding fiber contamination. Secondly, a PFBHA derivatization solution (87 mg of PFBHA in 2 mL pure) was prepared and located in a reagent tray position 1. Thirdly, PFBHA headspace vapor was generated in an incubation cell under optimized conditions of temperature, agitation time and speed, and adsorption time. Fourthly, SPME doped with PFBHA was located in the sampling cell where air is passed

through during a selected sampling time. Fifthly, pure MSTFA plus catalyst (40 µL plus 10 µL of TMCs) was prepared and located in a reagent tray position 2. Silylant reagent head space vapor was also generated in an incubation cell under





temperature, agitation time and speed, and adsorption time optimized conditions. Finally, the sample was analyzed by GC-MS. The chemical reactions involved in these derivatization steps are shown in Figure SI.1.

### 2.3 Chromatographic conditions

The sample was thermally desorbed in a GC port of an Agilent GC-MS (Santa Clara, USA) equipped with a HP-5MS column of 30 m × 0.25 mm I.D × 0.25 mm film (Agilent, Santa Clara, USA). The chromatograph was programmed at 80 °C for 2 min, then ramped at a rate of 12 °C min-1 to 240 °C, 100 °C min-1 to 280 °C. The injection port was held at 250 °C and the transfer line from GC to MS was held at 300 °C. Samples were injected in splitless mode, using helium as carrier at flow of 1 mL min$^{-1}$. The EI-voltage was 70 eV, the ion source temperature was set at 200 °C and the quadrupole temperature was set at 100 °C. Full scan mode was used (m/z 50-650) to identify the most abundant ions of multi-oxygenated compounds. The selected ion chromatograms of the most abundant ions were used to quantify them.

### 2.4 Smog chamber facilities and reference methods

The on-line SPME-GC-MS optimization was carried out in the high-volume outdoor smog chambers EUPHORE (European PHOtoREactor) (Valencia, Spain). These chambers consist of two half-spherical fluoropolymeric bags, each one of 200 m$^3$, with integrated measuring systems for monitoring pressure, humidity, temperature, precursor species, and reaction products. Pressure, relative humidity and temperature were measured using a pressure sensor (Air-DB-VOC, Sirsa, Madrid, Spain) and a dew point hydrometer (TS-2, Walz, Effeltrich, Germany). Also, a serenius 50 Ozone Monitor (Ecotech, Melbourne, Australia) was also used (Borrás and Tortajada-Genaro, 2012).

The OVOCs compounds were determined by two techniques. First, samples for on-line SPME-GC-MS technique were taken during 6 h at sampling frequency of 20 min (10 L min-1, sampling time: 5 min). Second, a White-type mirror system (path length of 553.5 m) coupled to a FTIR with MCT detector (NICOLET 670, Thermo Scientific, USA) was used. Spectra were collected at 1 cm$^{-1}$ resolution by averaging 300 scans (sampling time: 5 min). The quantification was based on aldehydic C-H stretching for methylglyoxal, within the spectral region of 2750–3000 cm$^{-1}$ and C-C and CH$_2$ bands for isoprene, methyl vinyl ketone and methacrolein in 900 – 1047 cm$^{-1}$ by using ANIR software (Ródenas, 2008).

For the optimization of the on-line SPME-GC-MS methodology, methylglyoxal, glyoxal, glutaraldehyde, methyl vinyl ketone, methacrolein, benzaldehyde, glycoldehydem hydroxyacetone, succinic acid, 4-oxo-2-pentenal and E-butendial - selected carbonyl, α-dicarbonyl, hydroxyl-carbonyl compounds and carboxylic acids - were injected in the smog chamber using an impinger. A stream of hot air (> 150 °C) produced by a hot gun enabled their volatilization and the transference inside the chamber at a flow of 10 L min-1. Moreover, our technique was applied in an intercomparison campaign and in the monitoring of isoprene ozonolysis.

**Intercomparison study.** A large intercomparison campaign of oxygenated organic compounds measurements was held at EUPHORE from mid-May to 1st June 2018 (Muñoz et al., 2019, Ródenas et al, In preparation). Taking advantage of it, our proposed SPME-GC-MS plus derivatization technique was compared towards a number of instruments: four on-line



instruments together with one off-line analytical method. The on-line techniques were on-line SPME-GC-MS plus derivatization, FTIR, a proton transfer time of flight mass spectrometer (PTR-ToF-MS), and a selected-ion flow-tube mass spectrometry (SIFT-MS). The off-line techniques were DNPH cartridges analyzed by LC-MS. These techniques and methodologies were operated by University of York, University of Leeds, Forschungszentrum Jülich and Fundación CEAM.

A detailed summary of techniques and institutions is in Table SI.1

Measurement of OVOCs at unknown concentrations were done under different relative humidity conditions. After overnight cleaning, all instruments sampled backgrounds air for 1 h. After that, OVOCs were added into the simulation chamber. I.e, methacrolein, acetone, 2-butanone, hydroxyacetone, glycolaldehyde, formaldehyde, benzaldehyde, acetaldehyde, methyl vinyl ketone, glyoxal and methylglyoxal were added in a range of 40 – 60 ppbV. Samples were taken during 1 h followed by

the addition of water to increase the relative humidity up to 50%. Subsequently samples were taken during a 30 min period and the gas mixture was diluted within 1 h down to 50% of the initial concentration. This protocol was repeated twice.

**Ozonolysis of isoprene.** The ozonolysis experiment consisted in adding Isoprene, 220 ppbV - 160 µL - were fed to the photoreactor via heated air stream. $O_3$ was added at 23 ppbV min$^{-1}$ from an $O_2$ 5.0 of purity using a UV lamp, until reach 215 ppbV. CO was added as OH radical scavenger (230 ppm of CO, 5000 ppmV at 10L min$^{-1}$). Later, all the reactants were

mixed for 10 min and all the instrumentation and techniques took samples during almost 6 h throughout the experiment.

## 3. Results and discussion

### 3.1 Set-up experiments of on-fiber PFBHA derivatization

The first challenge was an effective on-fiber derivatization of the carbonyl compounds to oxime products based on a substitution reaction by PFBHA. For assay development, methylglyoxal was selected as model compound and a synthetic air

mixture was generated at the EUPHORE smog chamber at 50 ppbV (confirmed by FTIR). Complete factorial designs were performed to study the variable effect (Table 1).

Table 1. Optimization experiments of on-fiber PFBHA derivatization

| Step | Variable | Conditions Range | Selected value |
|---|---|---|---|
| Headspace generation | PFBHA concentration | 10 – 170 g L$^{-1}$ | 87 mg L$^{-1}$ |
| | Loading temperature | 20 – 50 ºC | 50 ºC |
| | Agitation time | 1– 10 min | 3 min |
| | Agitation speed | 200 – 500 rpm | 500 rpm |
| Reagent adsorption | Adsorption time | 1– 10 min | 4 min |
| On-fiber | Sampling flow | 5 – 20 L min$^{-1}$ | 10 L min$^{-1}$ |


| | | | |
|---|---|---|---|
| derivatization | Sampling time | 1 – 10 min | 5 min |
| | Incubation Time | 0 – 10 min | 0 min |
| Desorption | Time | 1-15 min | 10 min |
| | Temperature | 150 – 250ºC | 250ºC |

The generation of PFBHA reagent vapor in a headspace mode was studied, varying reagent amount, temperature, agitation time and speed. The selection criteria were (1) maximum oxime signal; (2) minimal artifacts for the reagent depletion; (3) minimal standard deviation between replicates. Results showed that the signal saturation was achieved at PFBHA solution of 87 mg L$^{-1}$. In fact, higher PFBHA concentrations reduced the derivatization capacity and increased the interfering peaks as it was previously described by Yu et al., 2017. Regarding the incubation temperature, the oxime formation lightly improved

when incubation temperature cell increased. The best values were reported at 50 ºC. The modification of agitation time and speed factors were tested also leaded an improvement of reaction yield and reproducibility. Once an effective generation of the derivatizing reagent in the headspace was achieved, the fiber exposition was evaluated. The chosen adsorption time was 4 min, in accordance with previous studies (Gómez-Alvarez et al., 2012).

The next experiments were aimed at the on-fiber derivatization of carbonyl compounds. Sampling flow of 10 L min$^{-1}$ during

5 min and no-incubation time produced maximum capture of analyte at fast sampling rate. Finally, derivatized compounds were desorbed and directly transferred to the GC-MS. Selected conditions (10 min at 250ºC) assured high efficiency, calculated from peak areas of methylglyoxal derivatives (13.9 min, 14.2 min, 14.4 min, 14.5 min) being negligible the underivatized peak (retention time 7.65 min). Thus, the on-line SPME PFBHA-derivatization was completed for up to 95% methylglyoxal in a reduced time (total time 18 min) and low reagent consumption (0.17 mg per assay).

The reusability of PDMS/DVB fibers was tested, performing replicate experiments (> 50 assays). Since reproducible responses were registered, on-fiber derivatization is a cost-effective approach for the detection of OVOCs in air samples.

**3.2 Determination of carbonyl and α-dicarbonyl compounds after on-fiber PFBHA derivatization**

The multiplexed automated on-fiber derivatization was approached, studying the methodology for a mixture of eight

carbonyl compounds, including aldehydes, ketones, aromatic aldehydes and α-dicarbonyls. Then, the air sample contained selected compounds: methylglyoxal, glyoxal, glutaraldehyde, methyl vinyl ketone, methacrolein, benzaldehyde, 4-oxo-2-pentenal and E-butendial.

The chromatograms obtained from air mixtures confirmed the correct on-fiber derivatization and later chromatographic separation process. The chromatographic oxime peaks of each carbonyl compounds and its corresponding mass spectra are

reported in Figure SI.2 and SI.3. The identification of derivatives was based on the molecular ion in the EI mass spectra. The m/z fragments were 265 for both methacrolein and methyl vinyl ketone; 301, 448, 462, 490, 488, and 476, for benzaldehyde, glyoxal, methylglyoxal, glutaraldehyde, 4-oxo-2-pentenal and, E-butendial, respectively. Also, several fragments were





examined, such as ions with m/z = M-30, M-181, and M-211, resulting from loss of NO, $C_6F_5CH_2$, $C_6F_5$-$CH_2NO$, respectively. Table 2 summarizes the main chromatographic features for the specific oximes formed.


Table 2. Description of chromatographic peaks detected after the on-fiber derivatization using PFBHA as reagent for the carbonyl and α-dicarbonyl compounds selected.

| r.t (min) | Compound | Main m/z fragments | r.t of other oxime peaks (min) |
|---|---|---|---|
| 7.2 | Methacrolein | 84,181,235,265 | 7.3 |
| 7.5 | Methyl vinyl ketone | 181,235,265 | 7.6 |
| 11.9 | Benzaldehyde | 90,120,181,271,301 | 12.0 |
| 14.0 | Glyoxal | 181,267,418,448 | 14.1 |
| 14.8 | Methylglyoxal | 181,251,432,462 | 13.8, 14.2, 14.7 |
| 15.8 | E-4-oxo-2-pentenal | 181,307,458,488 | 15.6, 15.7, 16.0, 16.1, 16.2, 16.3, 16.4 |
| 16.0 | Glutaraldehyde | 181,279,309,460,490 | - |
| 16.1 | E-butenedial | 181,293,474 | 15.9, 16.0, 16.2 |

r.t = retention time

The number of underivatized compounds was negligible (< 1.5 %) and the number of PFBHA interference residue peaks was
reduced and perfectly resolved (resolution > 1.5). The calculated recoveries ranged from 91% (glutaraldehyde) and 99.7% (methylglyoxal) as shown in Table SI. 2. Therefore, the suitability of the developed method was demonstrated for a wide range of compounds, considering organic functionalities, molecular sizes, boiling temperatures and reactivities. Therefore, our method avoided the main limitation of SPME-based methods for multiplexed purposes. Often, the simultaneous determination of several analyte families led to a wide range of recoveries, because these methods are highly sensitive to
interferences and experimental conditions, such as sampling time, analyte concentrations, passive or active systems, reagent concentration, temperature, operator handling, etc (Koziel and Novak 2002).

The effect of humidity was examined because several techniques such as PTRMS-TOF or CEAS showed an erroneous determination for air samples with high water content, depends on the applied data evaluation routine (Talman et al., 2015). Air mixtures of the model organic compounds were prepared and mixed with water vapor. Statistical test demonstrated that
the humidity effect was negligible (test t, p values < 0.005).

**3.3 Analytical performances of on-fiber PFBHA derivatization**

Air mixtures at different concentrations were analyzed concentration range from 5 pptV to 100 ppbV. As a selective separation was achieved, appropriate calibration curves were obtained. The regression coefficients were 0.990-0.998.





The evaluation of sensitivity was performed analyzing blank samples and serial dilutions of standard air mixtures (dilution factor 1:10). Detection limits (LOD) were calculated as 3 times the standard deviation of blank samples at the retention time corresponding of each compound (Table 3). The estimated detection limits ranged from 6 pptV (glyoxal) to 100 pptV (methacrolein), equivalent to 14 to 237 ng m$^{-3}$. Quantification limits were calculated as 10 times the standard deviation of blank samples at the retention time corresponding of each compound. The limits were from 20 pptV (glyoxal) to 330 pptV

(methacrolein). The LOD values were compared to those reported in previously published papers. For glyoxal, mist chamber (Cofer scrubber) obtained LOD of 2.7 pptV (Spaulding et al., 2002); DOAS system 2 pptV, CEAS systems 19 pptV (Pang et al., 2013) or 75 pptV in Pang et al., 2014. For methylglyoxal, other studies obtained LOD 170 pptV using CEAS systems (Pang et al., 2013), 185 pptV in Pang et al., 2014 or 89 pptV using PTRMS-ToF (Michoud et al., 2018), being our detection limit of 97 pptV. The detection limit of our technique is comparable with most spectrometric methods and improves the

values for small molecules such as glyoxal and methylgyoxal. This result is particularly relevant because these α-dicarbonyl compounds play a key role on SOA formation in the atmosphere. Although this research was performed in a high-volume simulation chamber (200 m$^3$), the detection limits were below to typical concentrations of these compounds observed in ambient air.

Method precision was estimated from replicate experiments at 25, 50 and 100 ppbV (n=5). The percentage of relative

standard deviation was 0.2 – 7%. Other techniques reported higher errors, in Pang et al., 2013, the error was 22% using microfluidic lab-on-a-chip derivatization methodology. On the other hand, in Pang et al., 2014 the reproducibility was 6.6% for glyoxal and 7.5% for methylglyoxal using the same methodology. Therefore, we can conclude that our method presented comparable, even better, results.

Table 3. Analytical performances for the determination of carbonyl and α-dicarbonyl compounds.

| Compound | Molecular Formula | Linear Range (ppbV) | L.D (pptV) | L.Q (pptV) | RSD (%) |
|---|---|---|---|---|---|
| Methacrolein | C$_4$H$_6$O | 5 - 50 | 100 | 300 | 7 |
| Methylvinylketone | C$_4$H$_6$O | 10 - 100 | 70 | 200 | 6 |
| Benzaldehyde | C$_7$H$_6$O | 5 - 100 | 8 | 30 | 3 |
| Glyoxal | C$_2$H$_2$O$_2$ | 1 - 25 | 6 | 20 | 0.2 |
| Methylglyoxal | C$_3$H$_4$O$_2$ | 5 - 50 | 9 | 300 | 3 |
| E-4-oxo-2-pentenal | C$_5$H$_6$O$_2$ | 0.9 - 20 | 9 | 30 | 1.5 |
| Glutaraldehyde | C$_5$H$_8$O$_2$ | 5 - 50 | 50 | 170 | 5 |
| E-Butenedial | C$_4$H$_4$O$_2$ | 0.05 - 20 | 50 | 150 | 2 |





### 3.4 Extension to on-fiber PFBHA plus MSTFA derivatization

The on-line derivatization of hydroxyl or carboxylic groups, directly or after the described process for carbonyl groups, was studied based on a silylation reaction. For that, MSTFA was chosen as silylant. Besides previously selected ketones and aldehydes, hydroxyacetone, glycolaldehyde and succinic acid were selected as hydroxyl-carbonyl and carboxylic acid model compounds, due to their atmospheric relevance (Pospisilova et al., 2020; Mellouki et al., 2015). The derivatization sequence studied implied adsorption of PFBHA, sample loading and MSTFA plus catalyst adsorption. Thus, the oximes (-C=O groups) were generated before the formation of silanes (-OH and -COOH groups) Alternatively, BSTFA was tested as a silylant reagent, but the chromatograms showed more residual peaks and lower response for product peaks. These data suggested that short chain or branched compounds were hindered by steric impediment, as it was observed in solution derivatization (Borrás and Tortajada-Genaro, 2012).

The main experimental variables of MSTFA-based method were optimized from the corresponding chromatographic peaks (Table SI.3). Together with the molecular ions and PFBHA-associated fragments, the main ions were m/z = M-15 and M-73 resulting from loss of $CH_3$ and $Si(CH_3)_3$. The results indicated that the direct loading of vaporized reagent and catalyst (TMCS) was an effective way to transform PFBHA-products formed on the PDMS/DVB fiber. The absence of memory column effects and low number of artefacts were observed. Also, a successful thermal desorption of PFBHA-MSTFA products was achieved given the quantitative recoveries (> 95%).

Regarding the analytical performances, the detection limit was 0.08 – 0.15 ppbV, the linear range was 5-150 ppbV and reproducibility, expressed as standard deviation, was 2-4 % (see Table 4). In conclusion, double derivatization treatment allowed the proper determination of OVOCs, independently of the functionalized group (-C=O, -OH and/or -COOH), even with α-hydrogen.

Table 4. Analytical performances for the determination of hydroxyl-carbonyl and carboxylic acid model compounds.

| Compound | Molecular Formula | Linear Range (ppbV) | L.D (pptV) | L.Q (pptV) | RSD (%) |
|---|---|---|---|---|---|
| Hydroxyacetone | $C_3H_6O_2$ | 5 - 150 | 150 | 400 | 4 |
| Glycolaldehyde | $C_2H_4O_2$ | 5 - 150 | 20 | 100 | 3 |
| Succinic acid | $C_4H_6O_4$ | 5 - 150 | 8 | 50 | 1 |

### 3.5 Validation in an intercomparison study

An intercomparison campaign for the measurement of small multi-oxygenated compounds was carried out at EUPHORE atmospheric simulator. The study included the variation of OVOCs concentration and the impact that the presence of potential interferants, such as high humidity, and dilution steps can induce in the methodologies evaluated (see Table SI.1).



From the different OVOCs, we selected methylglyoxal since was previously selected such as OVOC model (see section 3.1). Figure 2 shows the average concentration in each step. Some techniques (spectroscopic and off-line) presented great interferences. The methodologies provided consistent results of methylglyoxal quantification. As can be observed, the results

from SPME-GC-MS plus derivatization technique were in great agreement with the theoretical values - a known quantity of compound was introduced into EUPHORE chamber - and with the results obtained by other techniques, both optical and spectroscopic methodologies. In fact, we can affirm that it does not present interferences with relative humidity lower than 60%. For that, a test t for paired samples was performed with a result of α < 0.5 for our proposed methodology, confirming that high humidity did not significantly affect our measurements.

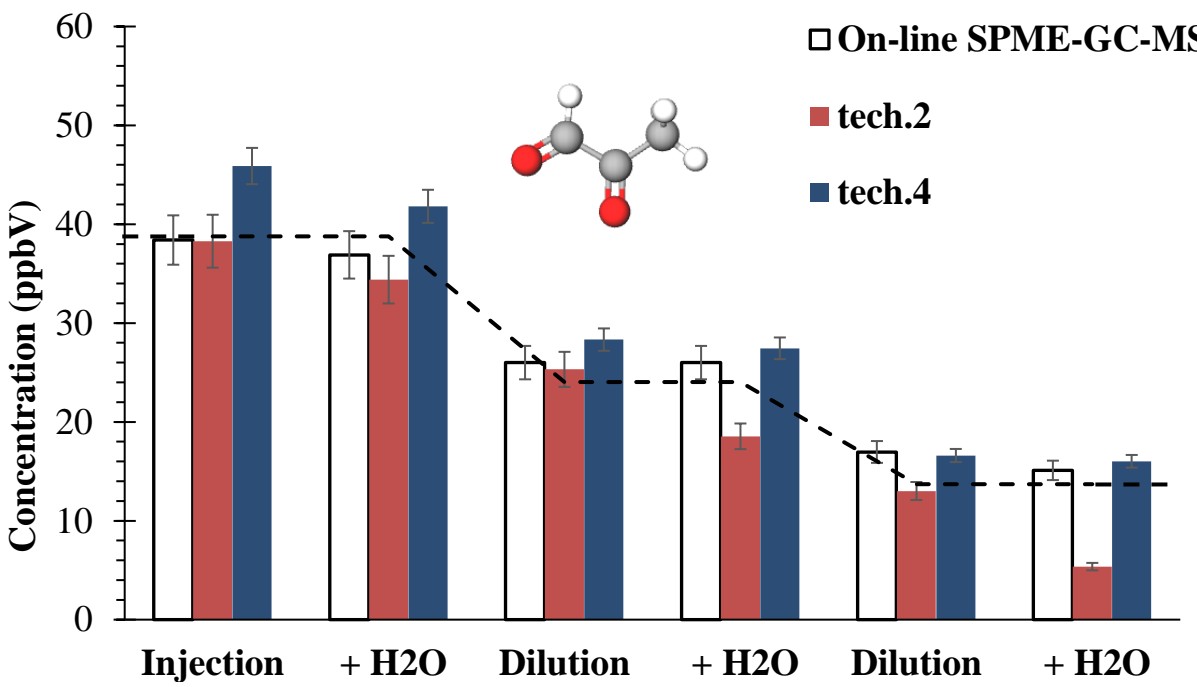


Fig. 2. Average concentration of methylglyoxal provided by on-line SPME-GC-MS, FTIR (tech.2) and PTR-ToF-MS (tech.4 from Jülich). The dashed line indicates the theoretical value, according to the quantity injected into the chamber, and the steps show this adjusted value with the dilution of the chamber (according to $SF_6$ values). Errors from SPME and FTIR include precision and accuracy; PTR-ToF-MS error consider 3σ of precision (did not include accuracy).


The intercomparison included OVOCs identified as main degradation products of biogenic pollutants (Aschmann and Atkinson, 1994; Iannone et al., 2010). Figure 3 shows the determined concentration levels of methacrolein and methyl vinyl ketone in a mixture containing both compounds. As previously observed, the results from SPME-GC-MS plus derivatization technique agreed with the theoretical values from chamber dilutions. Regarding other techniques, the concentrations were

comparable to SIFT-MS, FTIR and DNPH cartridges analyzed by LC-MS. However, PTR-ToF-MS cannot discriminate





between the structural isomers of both carbonyl compounds. In this the sum of MVK and MACR are measured due to PTR-MS methods are not selective. Both compounds have different sensitivity factor imposing an additional inaccuracy on the data, for more details see Ródenas et al., in preparation, 2021. On the contrary, on-line SPME-GC-MS approach can be used for a reliable monitoring of atmospheric reactions.


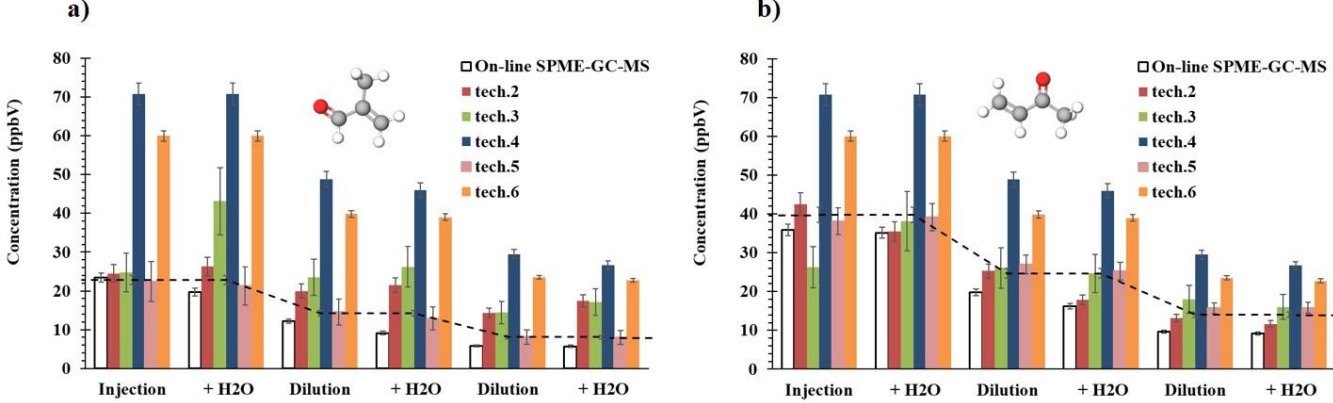

Fig. 3. Average concentration of methacrolein (top) and methyl vinyl ketone (bottom) for all compared techniques: On-line SPME-GC-MS, FTIR (tech.2), DNPH-LC-MS (tech.3), Ionicon-PTR-ToF-MS (tech.4), SIFT-MS (tech.5) and, KORE-PTR-ToF-MS (tech.6). Errors from SPME, FTIR and DNPH-LC-MS include precision and accuracy; Ionicon-PTR-ToF-MS,
SIFT-MS and KORE-PTR-ToF-MS errors consider $3\sigma$ of precision (did not include accuracy).

### 3.6 Application to monitoring of an ozonolysis reaction

The ozonolysis of isoprene, one of main biogenic compound emitted to the atmosphere, was studied at the EUPHORE
atmospheric simulator. On-line SPME-GC-MS plus PFBHA derivatization and MSTFA derivatization was applied for tracing the formation of the multi-oxygenated compounds. Figure 4a shows the decay of isoprene and ozone and the major degradation product (formaldehyde) formation. The degradation rate fitted to first order decay has been previously reported (Karl et al., 2004). Regarding to minority products, the OVOCs determined were 2-butanone, methacrolein, methyl vinyl ketone glycoladehyde, hydroxyacetone glyoxal and methylglyoxal. Some of them are plotted on Figure 4b. The results fitted
to a standard growing for degradation products. In case of 2-butanone, the formation was fast and, after 1 h, a further transformation was registered. The maximum concentrations were (23.6±1.2) ppbV, (25.4±1.3) ppbV, (4.2±0.2) ppbV, (4.1±0.2) ppbV, (3.8±0.2) ppbV (2.1±0.1) ppbV, and (2.2±0.1) ppbV, respectively. Other OVOCs such as methyl vinyl ketone were also detected. A total of seven OVOCs were identified and quantified in the isoprene ozonolysis, in good accordance with previous studies (Karl et al., 2004; Wennberg et al., 2018).






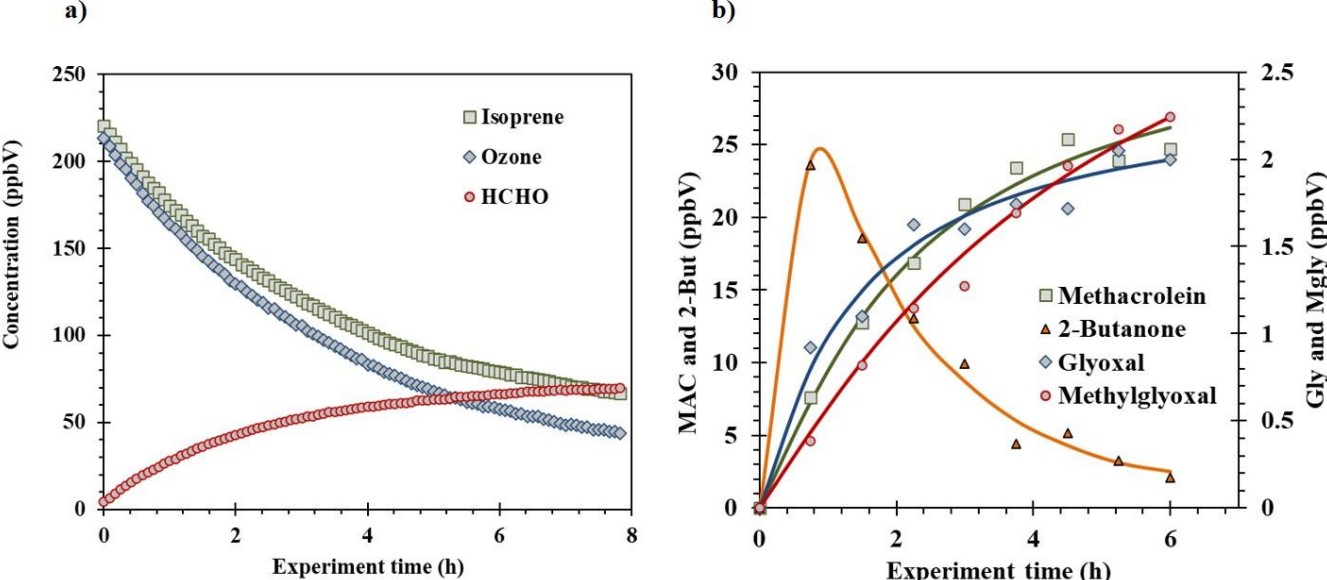

Fig. 4. Concentration profiles obtained from ozonolysis of isoprene at the high-volume atmospheric simulation chamber. Reagents and main products determined by FTIR (top). Main multi-oxygenated organic compounds determined by on-line SPME-GCMS (bottom). $[Isoprene]_0$ = 220 ppV; $[Ozone]_0$ = 215 ppbV; [CO scavenger] = 230 ppmV; Reaction volume = 200 m$^3$.

Therefore, our proposed on-line SPME-GC-MS plus derivatization enabled the accurate sensitive atmospheric monitoring of secondary pollutants. This technique will support their relevance since the ozonolysis of biogenic VOCs such as isoprene gives multifunctional oxygenated organic compounds that participate in the formation of aerosols. SOA formation during the atmospheric oxidation of biogenic organic compounds is estimated at 20-380 Tg r$^{-1}$ globally, influencing on human health and on climate (Mellouki et al., 2015).

## 4. Conclussions

The oxygenated volatile organic compounds play an important role in the atmosphere, even at low concentrations, so their reliable determination is challenging. On-line SPME-GC-MS double derivatization has demonstrated an efficient approach for alcohols, aldehydes, ketones, carboxylic acids and their combinations, independently on molecular size or structure. Compared to other techniques, such as FTIR, PTR-ToF-MS among others, this approach provides discontinuous data (20 min). However, on-line SPME-GC-MS method shows excellent analytical performances such as LOD (6-100 pptV), reproducibility (0.2-7%), selectivity (high resolution), cost-effectiveness and high-throughput. It is important to highlight the advantages compared to other chromatographic techniques, derived from an automated format, such as null solvent consumption, low reagent amounts, hands-off preparation, reusability of SPME fiber, robustness and high precision.





Demonstrated for the ozonolysis of isoprene, the number of potential applications is extraordinarily wide. On-line SPME-GC-MS double derivatization methodology could support a better understanding of atmospheric chemistry processes and an accurate monitoring of their atmospheric levels. This technology is particularly useful in atmospheric simulation chambers for air chemistry studies. A relevant example is the research of OVOCs as secondary organic aerosols (SOA) and ozone

precursors. This information should help a better assessment of their impact on human health and climate change. Moreover, after an adequate adaptation for mobile laboratories, the monitoring of atmospheric OVOCs levels is feasible.

Data availability. The experimental data will be available via the EUROCHAMP-2020(https://data.eurochamp.org/data-access/chamber-experiments/, data center as well as upon request to the contact author.

Special issue statement. This article is part of the special is-sue "Simulation chambers as tools in atmospheric research (AMT/ACP/GMD inter-journal SI).

### Acknowledges

This project/work has received funding from the European Union's Horizon 2020 research and innovation programme

through the EUROCHAMP-2020 Infrastructure Activity under grant agreement Nº 730997 and from IMAGINA- Prometeo project (PROMETEO/2019/110 from Generalitat Valenciana) F. CEAM is partly funded by the GVA. We especially want to thank Ralf Tillmann, Sergej Wedel and David Reimer, for the measurements made with PTRMS by Forschungszentrum Jülich (FZJ) institution.

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
