# Peer review of "On-line SPME derivatization for the sensitive determination of multi-oxygenated volatile compounds in air"

_Atmospheric Measurement Techniques, 2020_

## Referee Comment (RC1)

Title: On-line SPME derivatization for the sensitive determination of multi-oxygenated volatile compounds in air
Author(s): Esther Borrás, Luis A. Tortajada-Genaro, Milagros Ródenas, Teresa Vera, Thomas Speak, Paul Seakins, Marvin D. Shaw, Alastair C. Lewis, and Amalia Muñoz
MS No.: amt-2020-498
MS type: Research article
Iteration: Initial Submission
Special Issue: Simulation chambers as tools in atmospheric research (AMT/ACP/GMD inter-journal SI)

**General comments:**

The paper addresses innovative methodology related to measurement of relevant gaseous compounds in a simulated atmosphere. This is compliant with the very scope of AMT.

The paper is the sum of excellent quality work and I would recommend it as a valuable asset for researcher performing simulation chambers experiments.

The experimental work and the own contribution are well embedded in the frame of theoretical description of the analytical techniques used and suits fully the scope of the research. More, the description of the experiments is detailed sufficiently to allow duplication of the results. A clear structure makes the content readable and understandable.

The literature references are extensive but not in excess.

However, a few paragraphs need more attention due to overseen minor language errors. This is valid for figure captions as well. These are listed below in the section "technical corrections".

The supplementary material is supportive and complete the main body of the paper with concrete experimental data. However, at this stage the supplementary part needs corrections. They are listed below, in the "Technical corrections" section.

**Specific comments:**

Lines 149 – 151: *There is no indication for the reference values for FTIR absorption cross-sections used here. Could be possible to indicate the source of these values?*

*Or, if the quantification was based on the amount of compound introduced in the chamber, how was ensured that this occurred without loses?*

*Was the gas-phase composition for all analysed compounds constant over 6 hours?*

Line 195: *To what refers "no-incubation time"? According to figure 1, in an incubation cell occurs the doping of fibers with PFBHA. Please clearify.*

Line 210: *How good is the separation/determination of methacrolein and MVK in a mixture since they seem so alike? (m/z, retention times)*

Line 325: …In case of 2-butanone, the formation was fast and, after 1 h, a further transformation was registered.

*This suggest that previously another transformation took place?*

Lines 291-292: compound was introduced into EUPHORE chamber - and with the results obtained by other techniques, both optical and spectroscopic methodologies.

*I would suggest "both optical and mass spectroscopic methodologies". FTIR is a kind of spectroscopy, though.*

*Please verify the names of the compounds in the main body and supplementary material. There are some inconsistencies as the compounds are not named in the same way overall in the paper. More, in the Supplementary material the names, structures and mol masses are not always correct indicated (s. list in the Technical corrections section).*

**Technical corrections:**

**Main paper body:**

Line: 55 "lab-on-chip" *should be replaced by* "lab-on-a chip", *for consistency*.

Line 95: …E-butenedial…

Line 111: "Sampling line was of sulfinert® material and it was heated at 80°C to avoid losses of steaky OVOCs compounds."

*I am sure the intended word was "sticky". Please replace.*

Line 133: for 2 min, then ramped at a rate of 12 °C min$^{-1}$ to 240 °C, 100 °C min$^{-1}$ to 280 °C.

*I wonder if here is really a rate of 100 °C min$^{-1}$ and not 10 °C min$^{-1}$ ?*

Line 196-197: …Selected conditions (10 min at 250 °C) assured high efficiency, calculated from peak areas of methylglyoxal derivatives (13.9 min, 14.2 min, 14.4 min, 14.5 min) being negligible the underivatized peak (retention time 7.65 min).

*This sentence needs attention as it is unreadable.*

Line 206: the selected compounds

*Please add the article before "selected".*

Line 220-221: reduced and perfectly resolved (resolution > 1.5). The calculated recoveries ranged from 91% (glutaraldehyde) and 99.7% (methylglyoxal) as shown in Table SI. 2.

*Please replace "and" by "to".*

Line 227-228: The effect of humidity was examined because several techniques such as PTRMS-TOF or CEAS showed an erroneous determination for air samples with high water content, depends on the applied data evaluation routine (Talman et al., 2015).

*Please change into "that depends" or depending".*

Line 233: Air mixtures at different concentrations were analyzed concentration range from 5 pptV to 100 ppbV.

*This sentence needs attention as it is unreadable.*

Line 277-279: In conclusion, double derivatization treatment allowed the proper determination of OVOCs, independently of the functionalized group (-C=O, -OH and/or -COOH), even with α-hydrogen.

*This sentence needs attention as it seems incomplete.*

Line 286: potential interferants, such as high humidity, and dilution steps can induce in the methodologies evaluated (see Table SI.1).

*Please decide if after "evaluated" should come an "here" or "in this work". Otherwise "evaluated" should precede "methodologies".*

Line 288: From the different OVOCs, we selected methylglyoxal since was previously selected such as OVOC model (see section 3.1).

*I would suggest: From the different OVOCs, we selected methylglyoxal since was previously used as OVOC model (see section 3.1).*

Lines 306-309: … In this the sum of MVK and MACR are measured due to PTR- MS methods are not selective. Both compounds have different sensitivity factor imposing an additional inaccuracy on the data, for more details see Ródenas et al., in preparation, 2021. On the contrary, the on-line SPME-GC-MS approach can be used for a reliable monitoring of atmospheric reactions.

*I would kindly suggest the revision of this fragment as it is not easily readable.*

Figure 3: *Please replace "top" and "bottom" by "panel a" and "panel b", respectively.*

Figure 4: *Please replace "top" and "bottom" by "panel a" and "panel b", respectively.*

*As only HCHO is produced, the second sentence should read "Reagents and main product determined by FTIR (panel a)."*

Line 342: Please replace with "Conclusions"

Line 345: … for alcohols, aldehydes, ketones, carboxylic acids and their combinations,…

*Do you think that here "mixtures" could be more appropriate? If not, please ignore this comment.*

Line 360: …This article is part of the special is-sue…

*Please delete the hyphen.*

**SUPPLEMENTARY INFORMATION**

*Please correct the title of the:* SUPPLEMENTARY INFORMATION *part.*

Table SI.1: *Do you think that merging the cells in the column corresponding to CEAM foundation and CEAM, respectively, would make it easier to read?*

Figure SI.2. *Please correct the mol mass of methylglyoxal (72 instead of 74) and E-butenedial (84 instead of 86). Please draw the correct structures for glutaraldehyde and 4-oxo-2-pentenal.*

Figure SI.3.:

d) *The mass 54.1 is present in the MS spectra although is missing in the frame.*

i), j) *Please correct the mol mass of methyl glyoxal as 72.*

k), l) *The presented structure corresponds to succinaldehyde. To be glutaraldehyde needs one more C. Please correct it accordingly.*

m), n) *The presented structures correspond to 2-buten-2-methyldial, not 4-oxo-2-pentenal. More, the peaks in the chromatogram in panel m) are labelled overall as 4-oxo-2-pentanal. Please make the corrections according to the right compound.*

o), p) *Please correct the name as E-butenedial. Please correct the mol mass of E-butenedial as 84.*

---

## Referee Comment (RC2)

The authors present a study that investigates the application of online SPME-GC-MS for the atmospheric measurement of OVOCs with various functional groups. Therefore, reagents are adsorbed on a fiber and carbonyl functions are derivatized with PFBHA. In the next step hydroxyl and carboxyl groups are derivatized by MSTFA and TMCs. Subsequently, samples are desorbed and analyzed by GC-MS.

The paper presents tests and validation of the method for 11 different OVOCs (8 with carbonyl function and 3 with hydroxy or carboxyl group). The method was applied in the EUPHORE simulation chamber and results are compared to FTIR and PTR-TOF-MS measurements. For example, a good agreement was shown for methylglyoxal measurements. In addition, degradation products of the isoprene ozonolysis were determined. The authors demonstrate that the SPME technique can be used for atmospheric applications and the paper fits in the general scope of AMT.

Although I recommend that this paper be accepted for publication, I have several comments and suggestions that the authors should consider before finalizing this paper.

Specific comments:

What is the temperature of the sampling cell? Is any memory effect visible that depends on the history of experiments?

Page 7, line 166 and page 9, line 229: Which range of humidity was tested? How was the humidification done?

Page 10, line 249: The determination of the precision is described very briefly. Do I correctly understand that for every compound 5 measurements each were performed at reactant concentrations of 25, 50, and 100 ppbv? For some species that is outside of the linear range. Is the precision valid over the whole concentration range used in measurements?

Page 11, line 276: Are there any other methods/instruments (references?) to compare the given SPME performance?

Page 12: How did you calculate the dilution in the EUPHORE chamber? Was a tracer used?

Page 12, line 287: From the different OVOCs, we selected methylglyoxal since was previously selected such as OVOC model.

    This sentence sounds very odd and the meaning is not clear to me. Please rephrase.

Page 12, line 288: Here the authors state that other techniques suffer large interferences. Two sentences later, it is written: "As can be observed, the results from SPME-GC-MS plus derivatization technique were in great agreement with the theoretical values […] and with the results obtained by other techniques […]." Please specify which interferences and other techniques you are talking about.

Page 12, line 303: In Fig. 3 a) and b), SPME-GC-MS measurements do not agree with theoretical calculations within the stated uncertainty. It looks like measurements underestimate theoretical calculations up to 30%. That should be addressed in the manuscript.

Page 12, Fig. 2: The reader would benefit from simplified labels instead of looking them up in the caption. Exchange labels "tech.2" and "tech.4" by FTIR and PTR-ToF-MS, respectively.

Page 13, Fig. 3: See comment to Figure 2 for labels tech.2, tech.4, and so on. How do KORE- and Ionicon-PTR-ToF-MS correspond to FZJ and Leeds instruments listed in Table SI.1? Please use a uniform nomenclature.

Section 3.6 needs some attention. The content is not very clear and it needs a carful language check. See the following comments.

Page 13, line 324: The results fitted to a standard growing for degradation products.

The meaning of this sentence is not clear. Please rephrase.

Page 13, line 325: In case of 2-butanone, the formation was fast and, after 1 h, a further transformation was registered.

I don't think that transformation is the right word here. What you want to say is that 1) 2-butanone is formed and 2) after 1 h, the 2-butanone is consumed. Do you have an idea what causes the strong loss compared to the other measured VOCs?

Page 13, line 328: Which OVOCs were identified? Can you give some examples of how good the agreement is? Did you compare measured time series to a chemical model (which one?)?

Page 14, Fig. 4: What is the meaning of the solid lines in b)? Is a fitted function? Which type of?

Technical comments:

Page 3, line 43: Their tropospheric *range levels* are highly variable…

    better use: Their tropospheric concentrations…

Page 5, line 111: …losses of *steady* OVOCs.

    I think you mean "sticky"

Page 7, line 162: … proton transfer time of flight mass spectrometer (PTR-ToF-MS), …

    According to Table SI.1, two PTR-ToF-MS instruments were used.

Page 10, line 243: PTR-ToF-MS

Page 10, line 243: Michoud et al., 2018 is not listed in the references. Please check references.

Page 10, Table 3: Please use uniform names in text and tables. In table the authors use L.D., L.Q., and RSD. In the text, LOD is used for limit of detection. Abbreviations for L.Q. (quantification limit) and RSD (relative standard deviation) are not introduced. Same applies to Table 4.

Page 12, line 288: … presented *great* interferences.

    Replace by "large".

Page 12, line 293: … a test t…

    Remove "t"

Page 13, line 323: Regarding to minority products, the OVOCs determined were 2-butanone, methacrolein, methyl vinylketone glycoladehyde, hydroxyacetone glyoxal and methylglyoxal.

    I suggest replacing minority by minor. Please check for missing commas.

Page 13, line 326: The *maximum* concentrations were…

    I would suggest rephrasing "Measured 2-butanone concentrations were…

Page 14, Fig. 4: In the caption, do you mean "top" = a) and "bottom"= b) ?

---

## Author Comment (AC1)

**On-line SPME derivatization for the sensitive determination of multi-oxygenated volatile compounds in air**

Esther Borrás[1], Luis Antonio Tortajada-Genaro[2], Milagro Ródenas[1], Teresa Vera[1], Thomas Speak[3], Paul Seakins[3], Marvin D. Shaw[4], Alastair C. Lewis[4], Amalia Muñoz[1]

[1] Fundación Centro de Estudios Ambientales del Mediterráneo (CEAM), 46980 Paterna, Valencia, Spain
[2] Departamento de Química-Instituto IDM. Universitat Politècnica de València, 46022, Valencia, Spain.
[3] University of Leeds, LS2 9JT, Leeds, UK.
[4] National Centre for Atmospheric Science, University of York, York YO10 5DD, UK.

*Correspondence to*: Amalia Muñoz (amalia@ceam.es)

**COMMENTS REVIEWERS**

**REVIEWER: 1**

**General comments:**
*The paper addresses innovative methodology related to measurement of relevant gaseous compounds in a simulated atmosphere. This is compliant with the very scope of AMT.*
*The paper is the sum of excellent quality work and I would recommend it as a valuable asset for researcher performing simulation chambers experiments.*
*The experimental work and the own contribution are well embedded in the frame of theoretical description of the analytical techniques used and suits fully the scope of the research. More, the description of the experiments is detailed sufficiently to allow duplication of the results. A clear structure makes the content readable and understandable.*
*The literature references are extensive but not in excess.*
*However, a few paragraphs need more attention due to overseen minor language errors. This is valid for figure captions as well. These are listed below in the section "technical corrections".*
*The supplementary material is supportive and complete the main body of the paper with concrete experimental data. However, at this stage the supplementary part needs corrections.*
*They are listed below, in the "Technical corrections" section.*

RESPONSE: We would like to thank the reviewer for their comments and suggestions,

**Specific comments:**

Lines 149 – 151: *There is no indication for the reference values for FTIR absorption cross-sections used here. Could be possible to indicate the source of these values? Or, if the quantification was based on the amount of compound introduced in the chamber?*

RESPONSE: Thanks for your comment. The new paragraph is: "Second, a White-type mirror system (path length of 553.5 m) coupled to a FTIR with MCT detector (NICOLET 670, Thermo Scientific, USA) was used. Spectra were collected at 1 cm−1 resolution by averaging 300 scans (sampling time: 5 min). The quantification was based on aldehydic C-H stretching for methylglyoxal, within the spectral region of 2750–3000 cm−1 and C-C and $CH_2$ bands for isoprene, methyl vinyl ketone and methacrolein in 900 – 1047 cm-1 by using ANIR software (Ródenas, 2008). Calibrated references used in the analysis are found at the LAR database."

New reference has been included. Ref: LAR - Library of Analytical Resources: IR Spectra. Database of Atmospheric Simulation Chamber Studies of the EUROCHAMP-2020 Project. https://data.eurochamp.org/data-access/spectra.

*Was the gas-phase composition for all analysed compounds constant over 6 hours?*

RESPONSE: No. Concentrations of OVOCs compounds were corrected by dilution, an intrinsic process on simulation chambers. Reactants and products are diluted during experiments and to determine the correct concentration values, they must be corrected. The dilution rate in the chamber is calculated from the decay of $SF_6$ by FTIR areas in the IR range of 762-956 cm$^{-1}$. The specific dilution process was determined by FTIR adding 120 µg m$^{-3}$ of $SF_6$ as a non-reactive tracer (value of $1.1 \times 10^{-5}$ s$^{-1}$) to the reaction mixtures at the start of the experiments.

This information will be added to the manuscript

Line 195: *To what refers "no-incubation time"? According to figure 1, in an incubation cell occurs the doping of fibers with PFBHA. Please clearify.*

RESPONSE: Thank you for your comment. What we want to explain, and which is thus shown in Figure 1 is that the derivatizing reagent PFBHA does go through an incubation process in a cell at 50ºC but the sampling occurs in the sampling cell where the temperature is the same that the temperature of the simulation chamber.

Line 210: *How good is the separation/determination of methacrolein and MVK in a mixture since they seem so alike? (m/z, retention times).*

RESPONSE: The separation is very suitable because the oximes product of derivatization with PFBHA have quite different retention times. Individual standards were prepared to identify the oximes of each compound (as shown in Table 2). Methacrolein has 2 oximes with rt 7.2 and 7.3 while MVK has 2 oximes at 7.5 and 7.6. The chromatographic peaks with the largest area and the most reproducible are

80 selected, which are the peak of rt 7.2 for methacrolein and rt 7.5 for MVK, with which the differentiation is adequate and allows a correct quantification.

*Line 325: ...In case of 2-butanone, the formation was fast and, after 1 h, a further transformation was registered. This suggest that previously another transformation took place?*

85

*RESPONSE:* Sorry for the inconvenience. It was a mistake. Sentence has been modified: In case of 2-butanone, the formation was fast and, after 1 h, a decay by chemical degradation was registered.

*Lines 291-292: compound was introduced into EUPHORE chamber - and with the results obtained by*
90 *other techniques, both optical and spectroscopic methodologies. I would suggest "both optical and mass spectroscopic methodologies". FTIR is a kind of spectroscopy, though.*

RESPONSE: Thanks. The sentence has been included in the revised version.

95 *Please verify the names of the compounds in the main body and supplementary material. There are some inconsistencies as the compounds are not named in the same way overall in the paper. More, in the Supplementary material the names, structures and mol masses are not always correct indicated (s. list in the Technical corrections section).*

100 *RESPONSE:* Thanks for the comment. We have checked names and nomenclature along the revised draft.

**Technical corrections:**

105 RESPONSE: We accept technical corrections listed below. They will be included in the revised version.

In addition specific responses to comment on line 133: 100 °C min-1 is correct. We have a fast GC-MS
The sentence in line 233 should be replaced by : It has been changed to OVOCs air mixtures at different
110 concentrations were analyzed - ranged from 5 pptV to 100 ppbV

The sentence in line 277-279 should be: In conclusion, double derivatization treatment allowed the proper determination of OVOCs, independently of the functionalized group (-C=O, -OH and/or -COOH), even carbonyl compounds with α-hydrogen

115
Line 345. No we want to mean multifunctional compounds

**Main paper body:**
*Line: 55 "lab-on-chip" should be replaced by "lab-on-a chip", for consistency.*
120 *Line 95: ...E-butenedial...*

*Line 111: "Sampling line was of sulfinertR material and it was heated at 80°C to avoid losses of steaky OVOCs compounds."*

*I am sure the intended word was "sticky". Please replace.*

*Line 133: for 2 min, then ramped at a rate of 12 °C min-1 to 240 °C, 100 °C min-1 to 280 °C.*

*I wonder if here is really a rate of 100 °C min-1 and not 10 °C min-1 ?*

*Line 196-197: ...Selected conditions (10 min at 250 °C) assured high efficiency, calculated from peak areas of methylglyoxal derivatives (13.9 min, 14.2 min, 14.4 min, 14.5 min) being negligible the underivatized peak (retention time 7.65 min).*

*This sentence needs attention as it is unreadable.*

*Line 206: the selected compounds*

*Please add the article before "selected".*

*Line 220-221: reduced and perfectly resolved (resolution > 1.5). The calculated recoveries ranged from 91% (glutaraldehyde) and 99.7% (methylglyoxal) as shown in Table SI. 2.*

*Please replace "and" by "to".*

*Line 227-228: The effect of humidity was examined because several techniques such as PTRMS-TOF or CEAS showed an erroneous determination for air samples with high water content, depends on the applied data evaluation routine (Talman et al., 2015).*

*Please change into "that depends" or depending".*

*Line 233: Air mixtures at different concentrations were analyzed concentration range from 5 pptV to 100 ppbV.*

*This sentence needs attention as it is unreadable.*

*Line 277-279: In conclusion, double derivatization treatment allowed the proper determination of OVOCs, independently of the functionalized group (-C=O, -OH and/or - COOH), even with α-hydrogen.*

*This sentence needs attention as it seems incomplete.*

*Line 286: potential interferants, such as high humidity, and dilution steps can induce in the methodologies evaluated (see Table SI.1).*

*Please decide if after "evaluated" should come an "here" or "in this work". Otherwise "evaluated" should precede "methodologies".*

*Line 288: From the different OVOCs, we selected methylglyoxal since was previously selected such as OVOC model (see section 3.1).*

*I would suggest: From the different OVOCs, we selected methylglyoxal since was previously used as OVOC model (see section 3.1).*

*Lines 306-309: ... In this the sum of MVK and MACR are measured due to PTR- MS methods are not selective. Both compounds have different sensitivity factor imposing an additional inaccuracy on the data, for more details see Rodenas et al., in preparation, 2021. On the contrary, the on-line SPME-GC-MS approach can be used for a reliable monitoring of atmospheric reactions.*

*I would kindly suggest the revision of this fragment as it is not easily readable.*

*Figure 3: Please replace "top" and "bottom" by "panel a" and "panel b", respectively.*

*Figure 4: Please replace "top" and "bottom" by "panel a" and "panel b", respectively. As only HCHO is produced, the second sentence should read "Reagents and main product determined by FTIR (panel a)."*

*Line 342: Please replace with "Conclusions"*

*Line 345: ... for alcohols, aldehydes, ketones, carboxylic acids and their* *combinations,…*
*Do you think that here "mixtures" could be more appropriate? If not, please ignore this comment.*

165 *Line 360: …This article is part of the special* *is-sue**…*
*Please delete the hyphen.*

170 **SUPPLEMENTARY INFORMATION**

RESPONSE: We accept all the changes suggested below. They will be changed in the revised manuscript.

175 *Please correct the title of the:* SUPPLEMENTARY INFORMATION *part.*
Table SI.1: *Do you think that merging the cells in the column corresponding to CEAM foundation and CEAM, respectively, would make it easier to read?*
Figure SI.2. *Please correct the mol mass of methylglyoxal (72 instead of 74) and E-butenedial (84 instead of 86). Please draw the correct structures for glutaraldehyde and 4- oxo-2-pentenal.*

180 Figure SI.3.:
d) *The mass 54.1 is present in the MS spectra although is missing in the frame.*
i), j) *Please correct the mol mass of methyl glyoxal as 72.*
k), l) *The presented structure corresponds to succinaldehyde. To be glutaraldehyde needs one more C. Please correct it accordingly.*

185 m), n) *The presented structures correspond to 2-buten-2-methyldial, not 4-oxo-2 pentenal.*
*More, the peaks in the chromatogram in panel m) are labelled overall as 4-oxo-2 pentanal. Please make the corrections according to the right compound.*
o), p) *Please correct the name as E-butenedial. Please correct the mol mass of E butenedial as 84.*

---

## Author Comment (AC2)

**On-line SPME derivatization for the sensitive determination of multi-oxygenated volatile compounds in air**

Esther Borrás[1], Luis Antonio Tortajada-Genaro[2], Milagro Ródenas[1], Teresa Vera[1], Thomas Speak[3], Paul Seakins[3], Marvin D. Shaw[4], Alastair C. Lewis[4], Amalia Muñoz[1]

[1] Fundación Centro de Estudios Ambientales del Mediterráneo (CEAM), 46980 Paterna, Valencia, Spain
[2] Departamento de Química-Instituto IDM. Universitat Politècnica de València, 46022, Valencia, Spain.
[3] University of Leeds, LS2 9JT, Leeds, UK.
[4] National Centre for Atmospheric Science, University of York, York YO10 5DD, UK.

*Correspondence to*: Amalia Muñoz (amalia@ceam.es)

**COMMENTS REVIEWERS**

**REVIEWER: 2**

**General comments:**

*The authors present a study that investigates the application of online SPME-GC-MS for the atmospheric measurement of OVOCs with various functional groups. Therefore, reagents are adsorbed on a fiber and carbonyl functions are derivatized with PFBHA. In the next step hydroxyl and carboxyl groups are derivatized by MSTFA and TMCs. Subsequently, samples are desorbed and analyzed by GC-MS.*
*The paper presents tests and validation of the method for 11 different OVOCs (8 with carbonyl function and 3 with hydroxy or carboxyl group). The method was applied in the EUPHORE simulation chamber and results are compared to FTIR and PTR-TOF-MS measurements. For example, a good agreement was shown for methylglyoxal measurements. In addition, degradation products of the isoprene ozonolysis were determined. The authors demonstrate that the SPME technique can be used for atmospheric applications and the paper fits in the general scope of AMT.*
*Although I recommend that this paper be accepted for publication, I have several comments and suggestions that the authors should consider before finalizing this paper.*

RESPONSE: We would like to thank the reviewer for their comments and suggestions,

**Specific comments:**

*What is the temperature of the sampling cell? Is any memory effect visible that depends on the history of experiments?*

RESPONSE: As was indicate in line 115, sampling cell was under laboratory conditions that implies 20ºC and ambient pressure. We have added this information in the manuscript.

Regarding to memory effect, it was never occurred since all day we start with a SPME blank, SPME and PFBHA blank and no compound was detected in these chromatograms. Also, during methodology development, a blank was done during experiment sequence and no compound was detected, confirming the absence of memory effect. A new short phrase has been included in the revised manuscript.

Line 201 "Also, memory effect on sampling cell was evaluated, including a blank derivatized sampled included on sampling sequence. No compounds were observed, confirming the absence of memory effect".

*Page 7, line 166 and page 9, line 229: Which range of humidity was tested? How was the humidification done?*

*RESPONSE:* Thanks for your comment. Range of humidity test has been now included (0-50% RH). Moreover, a briefly description of humidity addition has been included in line 155 of the revised manuscript.

"Mili-Q water was added by a spryer system (dried cleaned air at 2 bar) into EUPHORE chamber. Dew point system was used to confirm relative humidity values".

*Page 10, line 249: The determination of the precision is described very briefly. Do I correctly understand that for every compound 5 measurements each were performed at reactant concentrations of 25, 50, and 100 ppbv? For some species that is outside of the linear range. Is the precision valid over the whole concentration range used in measurements?*

REPONSE: Sorry for the inconvenience. It was a mistake. Precision was estimated from replicate experiments of 1, 10, 25, 50 and 100 ppbV. It was emended in the revised text.

*Page 11, line 276: Are there any other methods/instruments (references?) to compare the given SPME performance?*

*RESPONSE:* As has been commented in the abstract and throughout the document, the reference technique for comparing the performance of the SPME methodology has always been FTIR. However, certain limits of detection and quantification, as well as precision and linear range are far from its performance.

*Page 12: How did you calculate the dilution in the EUPHORE chamber? Was a tracer used?*

RESPONSE: A new paragraph has been included in the revised version (line 155) for clarifying. "The dilution rate in the chamber is calculated from the decay of $SF_6$ by FTIR areas in the IR range of 762-956 cm$^{-1}$. The specific dilution process was determined by FTIR adding 120 µg m$^{-3}$ of $SF_6$ as a non-reactive tracer (value of $1.1 \times 10^{-5}$ s$^{-1}$) to the reaction mixtures at the start of the experiments".

*Page 12, line 287: From the different OVOCs, we selected methylglyoxal since was previously selected such as OVOC model. This sentence sounds very odd and the meaning is not clear to me. Please rephrase.*

RESPONSE: We have rephrased it. "From the different OVOCs, we selected methylglyoxal since was previously used as OVOC model (see section 3.1)".

85

*Page 12, line 288: Here the authors state that other techniques suffer large interferences. Two sentences later, it is written: "As can be observed, the results from SPME-GC-MS plus derivatization technique were in great agreement with the theoretical values [...] and with the results obtained by other techniques [...]." Please specify which interferences and other techniques you are talking about.*

90    RESPONSE: Ok thanks. As discussed at the beginning of the section, the interferences are "high humidity, and dilution steps can induce in the methodologies evaluated in this work (see Table SI.1)". The interferences are described: relative humidity and dilution process. The techniques are those that appear in table SI1 that are PTR-ToF-MS and SIFT-MS -spectroscopic techniques – and DNPH cartridges analyzed by LC-MS – off-line technique-.

95

*Page 12, line 303: In Fig. 3 a) and b), SPME-GC-MS measurements do not agree with theoretical calculations within the stated uncertainty. It looks like measurements underestimate theoretical calculations up to 30%. That should be addressed in the manuscript.*

Sorry it was a mistake when theoretical concentration was plotted. The correct figure 3 are:

[Figure]

100

[Figure]

*Page 12, Fig. 2: The reader would benefit from simplified labels instead of looking them up in the caption.*
105  *Exchange labels "tech.2" and "tech.4" by FTIR and PTR-ToF-MS, respectively.*
*Page 13, Fig. 3: See comment to Figure 2 for labels tech.2, tech.4, and so on. How do KORE- and Ionicon-PTR-ToF-MS correspond to FZJ and Leeds instruments listed in Table SI.1? Please use a uniform nomenclature.*

RESPONSE: Thanks for your comments. It was emended in the revised manuscript and new
110  figures were done.

*Section 3.6 needs some attention. The content is not very clear and it needs a carful language check. See the following comments.*
*Page 13, line 324: The results fitted to a standard growing for degradation products. The meaning of this*
115  *sentence is not clear. Please rephrase.*

RESPONSE: Thanks. It was rewritten. "The degradation rate of isoprene and ozone was fitted to first order decay as previously described in Karl et al., 2004".

120  *Page 13, line 325: In case of 2-butanone, the formation was fast and, after 1 h, a further transformation was registered. I don't think that transformation is the right word here. What you want to say is that 1) 2- butanone is formed and 2) after 1 h, the 2-butanone is consumed. Do you have an idea what causes the strong loss compared to the other measured VOCs?*

It is true, we have changed this sentence. "In case of 2-butanone, the formation was fast and, after
125   1 h, a decay by chemical degradation was registered". However, we have not made a study atmospheric
degradation mechanism because it is beyond the objective of this work

.

*Page 13, line 328: Which OVOCs were identified? Can you give some examples of how good the*
130   *agreement is? Did you compare measured time series to a chemical model (which one?)?*
      RESPONSE: OVOC identified is included: methyl vinyl ketone. We compare our data with
provided by Karl et al., 2004; Wennberg et al., 2018 in the literature.

*Page 14, Fig. 4: What is the meaning of the solid lines in b)? Is a fitted function? Which type of?*
135       RESPONSE: It was only to clarify the growth of methacrolein, glyoxal and methylglyoxal and the
different behaviour of 2-butanone.

*Technical comments:*
140

REPSONSE: All technical comments list below have been accepted

*Page 3, line 43: Their tropospheric range levels are highly variable… better use: Their tropospheric*
*concentrations…*
145   *Page 5, line 111: …losses of steady OVOCs.I think you mean "sticky"*
*Page 7, line 162: … proton transfer time of flight mass spectrometer (PTR-ToF-MS), …*
*According to Table SI.1, two PTR-ToF-MS instruments were used.*
*Page 10, line 243: PTR-ToF-MS*
*Page 10, line 243: Michoud et al., 2018 is not listed in the references. Please check references.*
150   *Page 10, Table 3: Please use uniform names in text and tables. In table the authors use L.D., L.Q., and*
*RSD. In the text, LOD is used for limit of detection. Abbreviations for L.Q. (quantification limit) and RSD*
*(relative standard deviation) are not introduced. Same applies to Table 4.*
*Page 12, line 288: … presented great interferences. Replace by "large".*
*Page 12, line 293: … a test t…Remove "t".*
155   *Page 13, line 323: Regarding to minority products, the OVOCs determined were 2-*
*butanone,methacrolein, methyl vinylketone glycoladehyde, hydroxyacetone glyoxal and methylglyoxal. I*
*suggest replacing minority by minor. Please check for missing commas.*
*Page 13, line 326: The maximum concentrations were… I would suggest rephrasing "Measured 2-*
*butanone concentrations were…*
160   *Page 14, Fig. 4: In the caption, do you mean "top" = a) and "bottom"= b) ?*
      Sorry for these mistakes. All of them has been solved in the revised manuscript.